# Glucose–Thymidine Ratio as a Metabolism Index Using ^18^F-FDG and ^18^F-FLT PET Uptake as a Potential Imaging Biomarker for Evaluating Immune Checkpoint Inhibitor Therapy

**DOI:** 10.3390/ijms23169273

**Published:** 2022-08-17

**Authors:** Sera Oh, Hyewon Youn, Jin Chul Paeng, Young-Hwa Kim, Chul-Hee Lee, Hongyoon Choi, Keon Wook Kang, June-Key Chung, Gi Jeong Cheon

**Affiliations:** 1Department of Nuclear Medicine, Seoul National University College of Medicine, Seoul 03080, Korea; 2Department of Biomedical Sciences, Seoul National University Graduate School, Seoul 03080, Korea; 3Laboratory of Molecular Imaging and Therapy, Cancer Research Institute, Seoul National University College of Medicine, Seoul 03080, Korea; 4Cancer Imaging Center, Seoul National University Hospital, Seoul 03080, Korea; 5Bio-MAX Institute, Seoul National University, Seoul 03080, Korea

**Keywords:** ^18^F-FDG, ^18^F-FLT, metabolism index, glucose–thymidine ratio(GTR), immune checkpoint therapy

## Abstract

Immune checkpoint inhibitors (ICIs) are widely used in cancer immunotherapy, requiring effective methods for response monitoring. This study evaluated changes in ^18^F-2-fluoro-2-deoxy-D-glucose (FDG) and ^18^F-fluorothymidine (FLT) uptake by tumors following ICI treatment as potential imaging biomarkers in mice. Tumor uptakes of ^18^F-FDG and ^18^F-FLT were measured and compared between the ICI treatment and control groups. A combined imaging index of glucose–thymidine uptake ratio (GTR) was defined and compared between groups. In the ICI treatment group, tumor growth was effectively inhibited, and higher proportions of immune cells were observed. In the early phase, ^18^F-FDG uptake was higher in the treatment group, whereas ^18^F-FLT uptake was not different. There was no difference in ^18^F-FDG uptake between the two groups in the late phase. However, ^18^F-FLT uptake of the control group was markedly increased compared with the ICI treatment group. GTR was consistently higher in the ICI treatment group in the early and late phases. After ICI treatment, changes in tumor cell proliferation were observed with ^18^F-FLT, whereas ^18^F-FDG showed altered metabolism in both tumor and immune cells. A combination of ^18^F-FLT and ^18^F-FDG PET, such as GTR, is expected to serve as a potentially effective imaging biomarker for monitoring ICI treatment.

## 1. Introduction

Immunotherapy is a cancer treatment method in which the immune system of the body is stimulated to fight cancer. Several immunotherapies have been used, including immune checkpoint inhibitors (ICIs), lymphocyte-activating cytokines, T cells, cancer vaccines, and oncolytic viruses [1]. ICIs block immune escape and induce active inflammation against cancer cells by targeting immune checkpoint proteins, such as programmed cell death-1 (PD-1) and programmed cell death ligand-1 (PD-L1), and have recently been used as a strikingly effective treatment for several types of cancers. In particular, they have been used as the most extensively studied treatments for advanced melanoma and have been correlated with improved survival [2,3]. However, despite many reports on the dramatic effects of ICIs, cancer responses are limited to specific subpopulations in clinical practice [4]. Additionally, PD-L1 expression in tumor specimens has limited predictive power, making it difficult to predict patient response to ICIs [5]. Thus, there is a need for effective biomarkers for ICI treatment to predict or evaluate early response.

Nuclear imaging is expected to be a valuable biomarker for ICI treatment, as it can easily evaluate the whole body in a repeatable and non-invasive manner. Several studies on early evaluation of response to immunotherapy have used positron emission tomography (PET) for PD-L1 expression, T cells, or various anti-tumor enzyme activities [6,7]. Additionally, characteristics of responders in terms of cellular components of the tumor microenvironment have been analyzed using molecular imaging [8].

^18^F-2-fluoro-2-deoxy-D-glucose (FDG) PET is the most widely used molecular imaging modality for cancer diagnosis, as it shows increased glucose metabolism, one of the hallmarks of cancers [9,10]. However, the main challenge with ^18^F-FDG is that glucose metabolism is increased not only in tumors but also in activated immune cells. Consequently, ^18^F-FDG uptake by tumors may be variable because both cancer and activated immune cells influence ^18^F-FDG uptake. ^18^F-fluorothymidine (FLT), a thymidine analog, is used as a PET tracer for nucleic acid metabolism. ^18^F-FLT is more specific for proliferative cells, especially cancer cells, than ^18^F-FDG, and might be a specific radiotracer for monitoring the efficacy of ICI treatment [11]. Different patterns are expected in ^18^F-FDG and ^18^F-FLT PET images during ICI treatment depending on the inflammatory activity and tumor response.

In this study, we evaluated the potential of combined ^18^F-FDG and ^18^F-FLT PET as an imaging biomarker for monitoring ICI treatment. ^18^F-FDG and ^18^F-FLT PET were performed during the ICI treatment, and their treatment-related changes were determined. Cell components in the tumor microenvironment were also analyzed as determinants for glucose and nucleic acid metabolisms.

## 2. Results

### 2.1. PD-L1 Expression in Mouse Melanoma

We first characterized PD-L1 expression in the mouse melanoma cell line B16F10. Since the expression of PD-L1 increases when mouse macrophages are activated, Western blotting showed high expression of PD-L1 in B16F10 cells similar to LPS-treated macrophages (Appendix A).

### 2.2. Therapeutic Efficacy of ICI Treatment

The effect of ICI treatment was confirmed by injecting the anti-PD-L1 monoclonal antibody into a syngeneic mouse model inoculated with B16F10 (Figure 1A). ICI treatment was effective, and tumor growth in the ICI treatment group was significantly inhibited compared to that of the control group. After 10 days, a trend of decreased tumor volume was evident in the ICI treatment group. The tumor size was substantially smaller in the treated group than in the control group on day 14 (182 ± 128 vs. 464 ± 300 mm^3^, *p* = 0.0049; Figure 1B, Appendix A), suggesting that ICI treatment was effective. Thus, the results of ICI treatment showed an anti-tumor effect in mouse melanoma, and differences in tumor growth resulting from anti-tumor immune responses.

### 2.3. PET Image Characteristics Associated with ICI Treatment

We next acquired PET images to predict the effect of ICI treatment in mouse melanoma model. ^18^F-FDG and ^18^F-FLT PET images were obtained in the early (day 6, 7) and late (day 13, 14) phases of treatment to analyze the therapeutic efficacy (Figure 2A). ^18^F-FDG PET in the early phase showed higher target-to-background ratio (TBR) in the ICI treatment group than in the control group (7.64 ± 2.02 vs. 5.16 ± 3.08, *p* = 0.0089; Figure 2B), whereas no difference in TBR was observed on the early ^18^F-FLT PET images between the two groups (6.75 ± 6.44 in the treatment group vs. 5.59 ± 4.07 in the control group, *p* = 0.4923; Figure 2C). In contrast, late ^18^F-FDG PET images exhibited no difference in TBR between the control (8.86 ± 1.90) and treatment groups (8.87 ± 2.91) (*p* = 0.9832; Figure 2D). However, the TBR of late ^18^F-FLT PET was markedly increased in the control group (13.79 ± 5.79) compared with the treatment group (4.32 ± 1.75, *p* < 0.0001) (Figure 2E). The sagittal and transverse images of ^18^F-FDG and ^18^F-FLT in the early and late phases of ICI treatment showed similar results (Appendix A).

Based on the imaging results, the glucose–thymidine uptake ratio (GTR) was defined as an index demonstrating the metabolic characteristics of the tumor, calculated as the TBR ratio of ^18^F-FDG PET and ^18^F-FLT PET. GTR was significantly higher in the ICI treatment group than in the control group on both early (1.90 ± 1.32 vs. 0.85 ± 0.60, *p* = 0.0035) (Figure 2F) and late (1.95 ± 0.93 vs. 0.72 ± 0.43, *p* < 0.0001 PET images (Figure 2G). These results indicated a relatively higher glucose metabolism than tumor proliferation at any time point in the ICI treatment group.

### 2.4. Comparison of Tumor Growth and Therapeutic Efficacy According to GTR in the ICI Treatment Group

Since we observed a large variation in tumor growth in the ICI treatment group (Figure 1B), further analysis was performed in terms of association between GTR and tumor growth.

ICI treatment group was divided into high and low GTR groups, by a cutoff of 1.50 on the early-phase images. In the high GTR group, the tumor growth (difference in tumor size between baseline and the endpoint) was 85.08 ± 49.24 mm^3^, whereas it was 173.88 ± 103.25 mm^3^ in the low GTR group (*p* = 0.0310, Figure 3A), indicating that the tumor growth was more inhibited in the high GTR group.

Additionally, ICI treatment group was divided into responder and non-responder by the tumor size on day 14, with a cutoff of 200 mm^3^ (the mean value). The responder group showed tendencies of higher TBR on ^18^F-FDG and lower TBR on ^18^F-FLT PET than non-responder group (Figure 3B,C), although they were not statistically significant.

### 2.5. Effect of ICI Treatment on Tumor Cell Components

Cell component analyses performed on days 7, 10, and 14 exhibited the most prominent difference between the ICI treatment and control groups on day 10. The proportion of total leukocytes (CD45+) was significantly higher in the ICI treatment group than the control group by flow cytometry on day 10 (5.07 ± 1.30% vs. 1.75 ± 0.36%, *p* < 0.0001) (Figure 4B–G). The pan-T (CD3+), helper (CD4+), cytotoxic (CD8+) T cells and macrophages (F4/80+) were increased in the treatment group compared to the control group (Figure 4G). Notably, the increase in cytotoxic T cell components was prominent. IHC staining also showed similar results. Tumor tissues of the ICI treatment group obtained on day 10 showed higher leukocyte infiltration (Figure 5). In particular, the increase in CD3+ and F4/80+ cells was prominent.

## 3. Discussion

In this study, we used ^18^F-FDG and ^18^F-FLT PET to evaluate glucose and nucleic acid metabolisms of tumors and their alterations following ICI treatment. ^18^F-FDG uptake was markedly increased in the early phase of ICI treatment and relatively decreased in the late phase. In contrast, ^18^F-FLT uptake showed a gradual decrease over time. Consequently, GTR, an index for glucose metabolism relative to nucleic acid metabolism, was consistently higher in the treatment group than the control group in both phases.

ICIs have been very successful in treating cancers such as melanoma, and represent a new paradigm in tumor treatment. However, ICIs also have several drawbacks, including high cost and adverse effects, such as autoimmune inflammation [12,13]. Moreover, responses are limited to specific subpopulations [14,15]. Therefore, to decide whether to administer or continue ICI treatment, it is necessary to select patients before treatment who can benefit from ICIs or to early evaluate the treatment efficacy [16]. Molecular imaging is a sensitive and mechanism-specific evaluation tool and is effective for tumor characterization and response monitoring, as well as tumor detection and localization. It is also useful for non-invasive imaging of immune responses [17,18,19,20,21]. Several studies have used molecular imaging to monitor cancer immunotherapy by targeting immune checkpoint proteins or specific cells in the tumor microenvironment [22,23,24].

^18^F-FDG is a glucose analog and is the most widely used PET imaging agent in clinical oncology [25]. Tumors metabolize ^18^F-FDG predominantly through the “Warburg effect”. However, ^18^F-FDG is not tumor-specific and also taken up by inflammatory cells, often leading to false-positive findings [10]. In tumor tissues, tumor-infiltrating macrophages can be a significant source of high ^18^F-FDG uptake, which is a limitation of ^18^F-FDG PET when used for response monitoring in ICI treatment. However, ^18^F-FLT, a radiolabeled nucleoside used for molecular imaging of nucleic acid metabolism, is mainly taken up by proliferating cells and incorporated into newly synthesized DNA. In vitro and in vivo animal studies have reported that ^18^F-FLT uptake is more specific for tumors than inflammatory lesions [26]. Therefore, we hypothesized that ^18^F-FLT would be better than ^18^F-FDG in the response monitoring of ICI treatment.

We used B16F10 mouse melanoma cells, known to highly express PD-L1 and generate immunologically “cold” tumors, exhibiting a variety of therapeutic responses [6,7,27,28]. With ICI treatment, tumor growth was effectively inhibited by inducing an immune reaction against the tumor. We observed overall tumor suppression after ICI treatment, and our study showed that the level of tumor suppression could be divided into two groups roughly on day 10. Therefore, we obtained tumor images at the early (day 6, 7) and late (day 13, 14) phases, and investigated cellular components in tumors with FACS and IHC at three time points.

In the early phase of ICI treatment, ^18^F-FLT uptake was not significantly different between the treated and control groups, suggesting a latent period for overt growth inhibition. However, ^18^F-FDG uptake was higher in the ICI treatment group even in the early phase, probably due to inflammation in the tumor. Cell component analysis of the tumor tissue demonstrated microenvironment changes corresponding to the ICI treatment. During the early phase, the proportions of T cells (CD3+), cytotoxic T cells (CD8+), and macrophages were slightly higher in the treatment group than the control group, although they were not statistically significant. It is possible that the ICI treatment activated immune cells, resulting in high glucose metabolism. On day 10, the proportion of tumor-infiltrating cells was altered, demonstrating an active immune reaction. Infiltrations of total leukocytes (CD45+) and T cells (CD3+) were significantly higher in the ICI treatment group than the control group. In the late phase, tumor growth was inhibited and ^18^F-FLT uptake was decreased in the ICI treatment group, whereas ^18^F-FLT uptake was markedly increased in the control group due to tumor progression. In contrast, ^18^F-FDG uptake was still high in the ICI treatment group probably due to combined active immune reaction.

^18^F-FDG appeared to be a marker for the metabolic activity of both tumor and immune cells, whereas ^18^F-FLT was a relatively specific marker for the proliferative activity of tumor cells. During ICI treatment, ^18^F-FDG uptake could be increased by activated immune cells regardless of tumor response. In clinical practice using ICI, pseudoprogression observed on ^18^F-FDG PET has been reported [9], which was probably related to enhanced glucose metabolism of immune cells. In contrast, ^18^F-FLT was relatively specific to tumor proliferation, but it was usually less sensitive to changes after treatment [26]. Thus, the combination of ^18^F-FDG and ^18^F-FLT PET is expected to provide synergistic information on tumor response to ICI.

In the present study, we defined a quantitative index of GTR as the ratio of ^18^F-FDG and ^18^F-FLT uptakes. GTR is deemed to represent the specific glucose metabolism of immune cells because overall metabolic activity is normalized by tumor-specific proliferation activity. When a tumor exhibits high ^18^F-FDG uptake, a high GTR would suggest relatively high activity of immune cells, while a low GTR would suggest high activity of tumor cells. In our study, GTR was higher in the ICI treatment group than the control group at both early and late phases, which was consistent with the presumed active immune reaction. Because both ^18^F-FDG and ^18^F-FLT are clinically available radiotracers, GTR is expected to be an effective marker for monitoring response to ICI treatment in clinical practice. Our study warrants further clinical studies using ^18^F-FDG and ^18^F-FLT PET.

In our study, responders and non-responders were not clearly separated although we attempted to divide them according to the tumor size on day 14. It was probably due to wide variation of tumor responses and insufficient sample numbers for subgroup analysis. The characteristics of GTR in responders and non-responders need to be investigated in further studies.

In conclusion, ICI treatment effectively inhibited B16F10 melanoma tumor growth and induced an active immune response against tumor cells. ^18^F-FLT PET showed relatively specific changes in tumor cell proliferation activity, whereas ^18^F-FDG PET revealed increased glucose metabolism in both tumor and immune cells. A consistently high ^18^F-FDG uptake was observed in the early and late phases of ICI treatment, which could be a limitation of ^18^F-FDG PET in response monitoring. Therefore, a combination of ^18^F-FLT and ^18^F-FDG PET, such as GTR, could be a potentially effective imaging biomarker for monitoring ICI treatment.

## 4. Materials and Methods

### 4.1. Tumor Cells

Mouse melanoma cell line B16F10 was obtained from the Korean Cell Line Bank (KCLB, Seoul, Korea). The cells were grown in Dulbecco’s modified Eagle medium supplemented with 10% fetal bovine serum and 1% antibiotics at 37 °C in a humidified atmosphere containing 5% CO_2_.

### 4.2. PD-L1 Expression

PD-L1 expression in B16F10 cells was analyzed by isolating total proteins using RIPA buffer (50 mM Tris-HCl, pH 8.0 with 150 mM sodium chloride, 1% Igepal CA-630, 0.5% sodium deoxycholate and 0.1% sodium dodecyl sulfate). The lysates were loaded onto 10% SDS-polyacrylamide gels, blotted onto polyvinylidene difluoride membranes (PVDF, Millipore, Watford, UK) and subsequently blocked with 5% skim milk for 1 h at room temperature. The membranes were incubated overnight at 4 °C with primary antibodies targeting PD-L1 (BioXcell, Lebanon, PA, USA) and β-actin (Sigma Aldrich, St. Louis, MO, USA) and then probed with horseradish peroxidase-conjugated anti-rat IgG (Enzo Life Sciences, Seoul, Korea) or anti-mouse IgG (Cell Signaling Technology, Danvers, MA, USA). The signal intensity was measured using a chemiluminescence imaging system (Bio-Rad, Hercules, CA, USA). Activated macrophage cells (RAW264.7, treated with lipopolysaccharide) were used as a positive control.

### 4.3. Tumor Model

The B16F10 cells (1 × 10^6^) were inoculated in the subcutaneous tissue of the flank area of female C57BL/6 mice (6–8 weeks old). The body weight was monitored, and the tumor size was measured daily using a digital caliper. All experiments were approved by the Institutional Animal Care and Use Committee at Seoul National University Hospital (SNU-160519) and animals were maintained in the facility accredited by the AAALAC International (#001169) in accordance with the Guide for the Care and Use of Laboratory Animals 8th edition, NRC (2010).

### 4.4. ICI Treatment and PET Image Acquisition

The overall study design is displayed in Figure 1. When the tumor diameter was 2–3 mm, ICI treatment was started using daily intraperitoneal injections of rat anti-PD-L1 antibody (10 mg/kg, 100 μL, BioXcell, Lebanon, PA, USA). The injections started on day 5 after tumor inoculation and were repeated daily through day 14. Mice in the control group were injected with phosphate-buffered saline (PBS) following the same protocol. PET scans were performed in “early” and “late” phases twice for ^18^F-FDG and ^18^F-FLT treatment (n = 26) and control (n = 20) groups. ^18^F-FLT PET was performed on day 6 (early) and day 13 (late), and ^18^F-FDG PET was performed on day 7 (early), and day 14 (late).

PET images were obtained using a small-animal PET scanner (GENYSIS^4^, Sofie Bioscience, Dulles, VA, USA). Each mouse was maintained fasting for at least 12 h before radiotracer injection. Mice were anesthetized with 1.5% isoflurane, and the radiotracers (1.18 ± 0.22 MBq/100 μL, ^18^F-FDG and ^18^F-FLT) were intravenously injected into the tail vein. One hour after injection, a PET image was acquired for 5 min. After the final PET image acquisition, mice were sacrificed for harvesting tumor tissues.

### 4.5. PET Image Analysis

Quantitative analysis of PET images was performed using the shareware software package AMIDE (Source Forge, San Diego, CA, USA). The standard uptake value (SUV) of the tumor was measured by placing a three-dimensional volume of interest on the tumor. The SUV was calculated as radioactivity concentration divided by injected radioactivity and body weight. For comparison, the SUV of the reference tissue was also measured, and the target-to-background ratio (TBR) was calculated by dividing the maximal SUV of a tumor by the mean SUV of the reference tissue. The liver was used for ^18^F-FDG as the reference tissue, and the forelimb muscle was used for ^18^F-FLT PET. The TBR on each image was compared between the ICI treatment and control groups. The PET index of glucose–thymidine uptake ratio (GTR) was defined as the ratio of TBRs on ^18^F-FDG PET and ^18^F-FLT PET, deemed a relative activity of glucose metabolism compared to the proliferative activity.

### 4.6. Cell Component Analysis in Tumors

To analyze tumor cell components, additional animal tumor models of control and treatment groups were generated using the protocol described above. After tumor cell inoculation, mice were sacrificed on days 7, 10, and 14 (n = 2, 4, 4, respectively). The tumor tissues were harvested and analyzed by flow cytometry and IHC. For flow cytometry, single-cell suspension was obtained from a tumor specimen and was treated for 30 min at 4 °C with fluorochrome-conjugated antibodies; CD3, CD4, CD8, CD45, or F4/80 (Abcam, Cambridge, UK). Cell components were identified by expression of specific antigens, including pan-T cells (CD45+CD3+), helper T cells (CD45+CD3+CD4+), cytotoxic T cells (CD45+CD3+CD8+) and macrophages (CD45+F4/80+). IHC staining was also performed using antibodies against CD3, CD4, CD8, granzyme B, F4/80, GLUT1, Ki67 (Abcam, Cambridge, UK). Tumor tissues were fixed in 10% paraformaldehyde at 4 °C and incubated with the antibodies overnight at 4 °C. After washing with PBS, the tissues were incubated with the secondary anti-rabbit antibody for 1 h at room temperature, incubated with the avidin–biotin complex for 1 h at room temperature, and developed with DAB.

Welch’s *t*-test was used for the statistical comparison of values between groups.

## Figures and Tables

**Figure 1 ijms-23-09273-f001:**
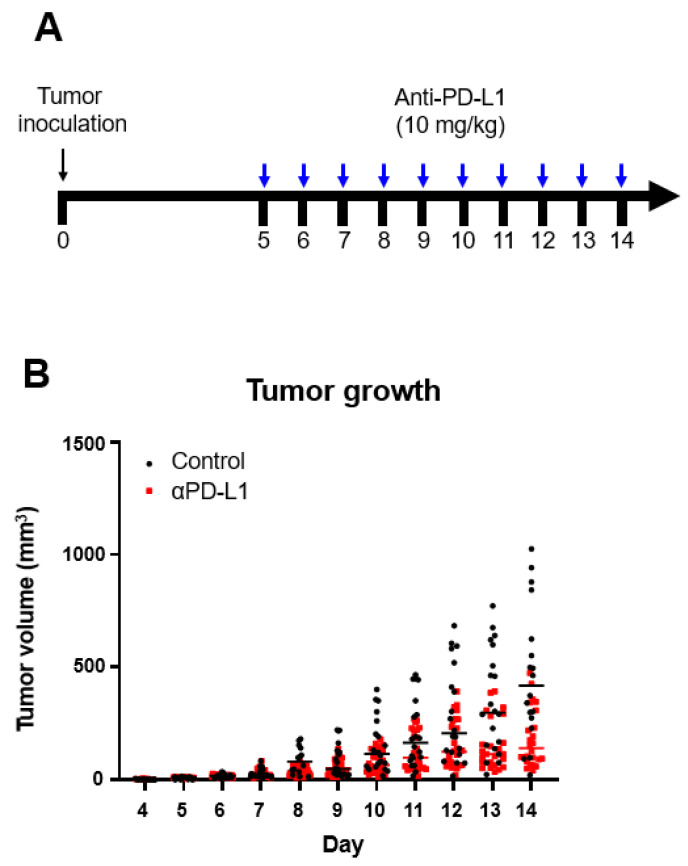
Anti-tumor effect of PD-L1 immunotherapy in a mouse melanoma model. (**A**) Study design. Murine melanoma B16F10 cells were inoculated into mice on day 0. Anti-PD-L1 antibody was injected daily starting from day 5 when tumor diameter was 2–3 mm. PET images were obtained at the early (days 6 and 7) and late (days 13 and 14) phases of treatment. (**B**) Tumor growth after ICI treatment in all groups.

**Figure 2 ijms-23-09273-f002:**
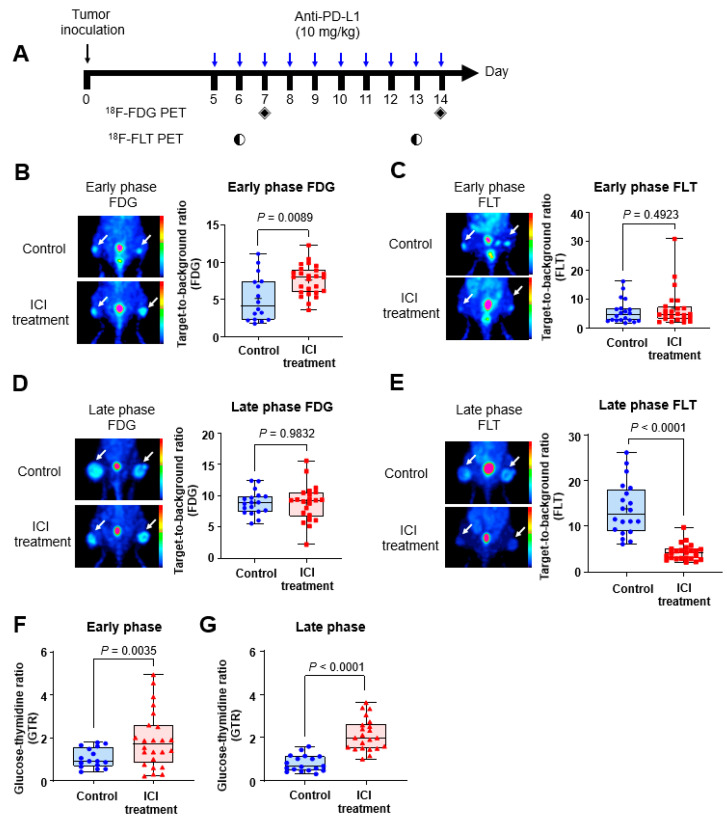
PET images in the early and late phases of ICI treatment. (**A**) Experimental design. ^18^F-FDG and ^18^F-FLT images were acquired in the early phases of ICI treatment (day 6, 7), and FDG and FLT images were acquired in the late phases of ICI treatment (day 13, 14). The black diamond symbols mean the day on which ^18^F-FDG PET images were obtained, and the black circle symbols mean the day on which the ^18^F-FLT PET images was acquired. The white arrow indicates the tumor uptake lesion. ^18^F-FDG PET showed high TBR in the ICI treatment group in the early phase (**B**), whereas TBR by ^18^F-FLT PET showed no difference in the early phase (**C**). In the late phase, there was no difference in TBR between the ICI treatment and control groups by ^18^F-FDG PET (**D**), whereas ^18^F-FLT PET showed a significantly low TBR in the treatment group (**E**). GTR was significantly higher in the ICI treatment group than the control group in both the early (**F**) and late (**G**) phases. In the graphs, the blue and red symbols represent data of each individual, and the black squares represent the data distribution.

**Figure 3 ijms-23-09273-f003:**
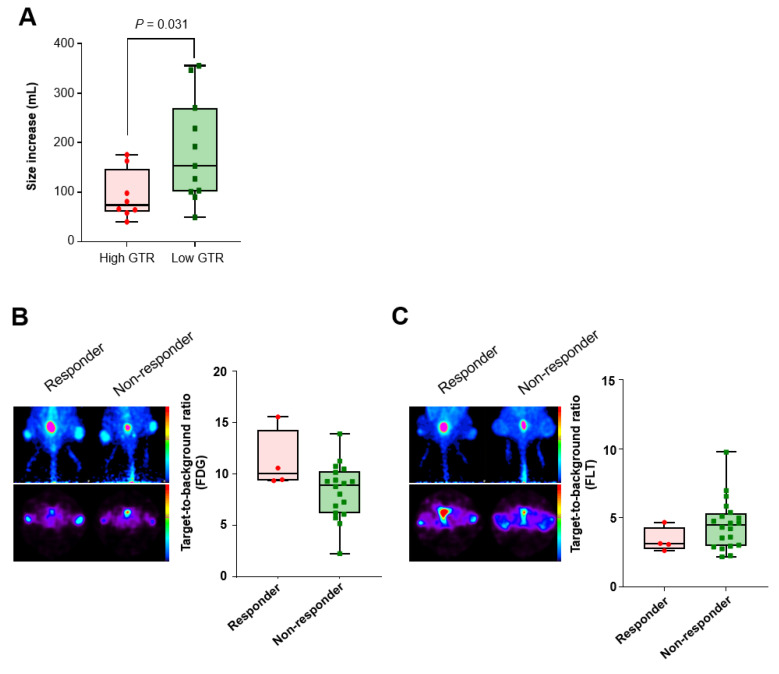
Comparison of tumor growth according to the image findings. (**A**) When the ICI treatment group was classified by the GTR with a cutoff value of 1.50, high-GTR tumors showed significantly lower growth than low-GTR tumors (*p* = 0.0310). On day 14, TBR on ^18^F-FDG PET images was relatively higher in responders than non-responders (**B**), whereas TBR on ^18^F-FLT PET images was relatively lower in responders than non-responders (**C**).

**Figure 4 ijms-23-09273-f004:**
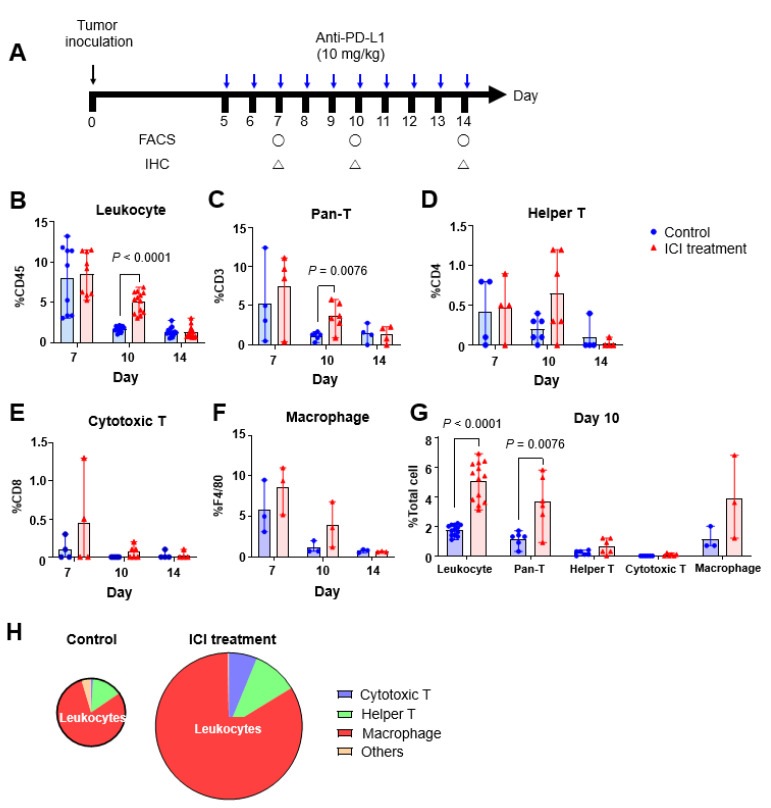
Cell components in tumors following ICI treatment. (**A**) Experimental scheme. After tumor inoculation, cells were isolated and analyzed at each phase of treatment. The black circle symbols mean the day on which the FACS experiment was performed, and the black triangle symbols indicate the day on which the IHC experiment was performed. (**B**) Percentage of total leukocytes (CD45+) was significantly increased in the ICI treatment group on day 10. (**C**–**F**) The ratios of leukocyte (CD45+) and pan-T (CD3+) cells were significantly higher in the ICI treatment group, and (**G**) Total leukocyte cell population showed higher immune cell infiltration in the ICI treatment group. Pan-T cells (CD3+), helper T cells (CD4+) and macrophages (F4/80+) showed a significantly higher percentage on day 10, indicating higher immune cell infiltration in tumors treated with ICI. (**H**) Absolute number of immune cells was also higher in the ICI treatment group. The *y*-axis of (**B**–**F**) means the count percentage of each cell markers in total cells population from tumor.

**Figure 5 ijms-23-09273-f005:**
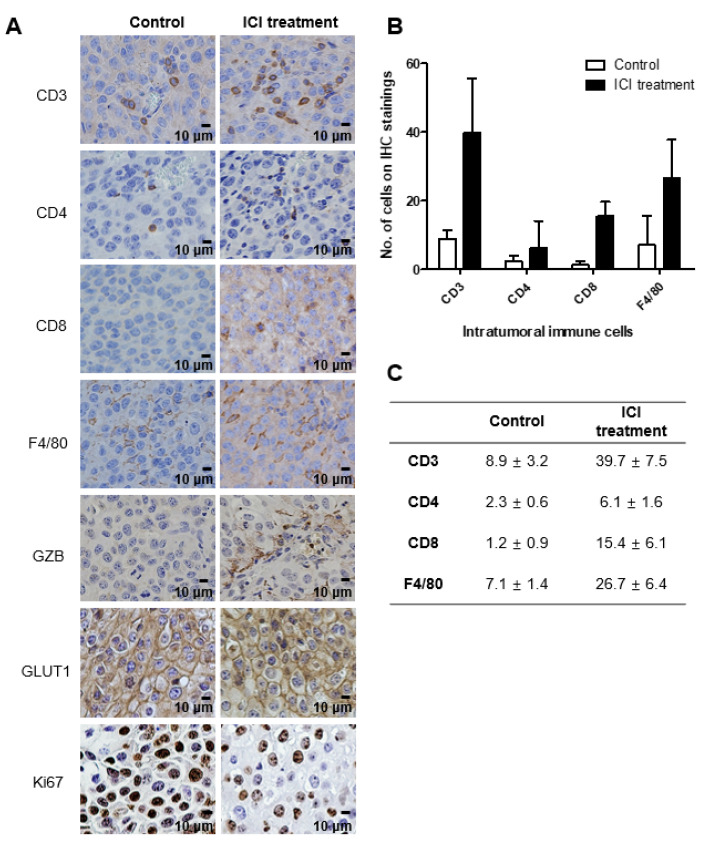
Histological analysis after ICI treatment. Intra-tumor infiltration of immune cells were observed in the ICI-treated group at Day 10. (**A**) Pan-T (CD3+), helper T cells (CD4+), cytotoxic T cells (CD8+), macrophages (F4/80+), granzyme B (GZB), glucose transporter (Glut-1), and cell proliferation (Ki-67) markers were stained. (**B**) The number of various immune cells infiltrated in the tumor tissue. (**C**) The number of intra-tumoral immune cells on IHC samples.

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
