# Peer review of "Glucose–Thymidine Ratio as a Metabolism Index Using 18F-FDG and 18F-FLT PET Uptake as a Potential Imaging Biomarker for Evaluating Immune Checkpoint Inhibitor Therapy"

_ijms, 2022, doi:10.3390/ijms23169273_

Round 1
Reviewer 1 Report
In this study Oh et al. investigate the use of 18F-FDG and 18F-FLT PET for evaluation of ICI therapy efficacy. The authors determine that the Glucose-Thymidine Ratio can predict the response to ICI therapy. This is an interesting study with strong clinical implication, but the use of only one model (B16F10) limits the significance of the finding. The study would be strengthened if authors show a similar effect in another responding cancer model and proof its validity by also measuring a non-responding cancer model where no effect should be observed. I am also wondering how authors think this ratio can be translated into clinical use where no control group is available?
Specific comments:
Figure 4:
- What is the % on the y-axis in B-F referring to? For example % CD3 out of what population? As CD3 is a pan-T cell marker shouldn’t CD4 and CD8 T-cells roughly add up to % CD3 (day 7 control – CD3 5 %, CD4 0.4%, CD8 0.1%)?
- Figure 5:
- Please count number of positive cells (CD3, CD4, CD8 etc) on IHC stainings from several mice and show a bar diagram for every marker next to the picture. The chosen IHC area is small and the reader relies on the image selection done by the authors. An unbiased readout by cell count would strengthen the point. Please also include an apoptosis marker such as cleaved Caspase3 or similar to prove the effectiveness of ICI therapy in these mice.
- In the material and methods authors mention use of granzyme B in IHC staining but shown is GLUT1. Please clarify.
Author Response
In this study Oh et al. investigate the use of 18F-FDG and 18F-FLT PET for evaluation of ICI therapy efficacy. The authors determine that the Glucose-Thymidine Ratio can predict the response to ICI therapy. This is an interesting study with strong clinical implication, but the use of only one model (B16F10) limits the significance of the finding.
The study would be strengthened if authors show a similar effect in another responding cancer model and proof its validity by also measuring a non-responding cancer model where no effect should be observed.
RE: Thank you very much for your comments. We agreed with the reviewer’s suggestion. However, there have been many reports of confirming ICI therapy in tumor models using immunologically inflamed tumors such as CT26 or MC38 [5, 21]. In addition, it has been reported that the response of paclitaxel (PTX) monotherapy was assessed using 18F-FDG and 18F-FLT in a triple negative breast cancer model [27]. According to the reviewer's comments, if it is verified by comparing it with another responding cancer model, the clinical significance will be further strengthened. However, we used B16F10 melanoma, known as a poorly immunogenic with variable ICI response, and showed a statistically tumor suppression compared to the control group. It means that ICI treatment was effective but it showed individual-variation. Therefore, we showed responding and less-responding tumors in this model. In addition, we cannot establish a new model with different cancer cell lines for revision because revision due date is only two weeks.
I am also wondering how authors think this ratio can be translated into clinical use where no control group is available?
RE: F-18-fluoro-deoxyglucose (18F-FDG) is the most widely applied clinical positron imaging agent. Also, F-18-fluoro-3'-deoxy-3'-L-fluorothymidine (18F-FLT) is a positron emission tomography tracer which reflects proliferative activity especially for the early evaluation of treatment response in many cancer patients. Quantitative evaluation of PET images is the most common clinical approach for measuring 18F-FDG or 18F-FLT uptake levels. It is known that quantitative data of PET images are analyzed in the form of the standardized uptake value (SUV) or standardized uptake ratio (SUR, target/reference) [28-29]. These concepts have been used for semi quantitative analysis based on the pharmacokinetic modeling with normal volunteers and patients. Since it is not possible to scan a normal control group whenever a patient's PET scan is performed, it has been recommended to evaluate by applying SUV and SUVR by various prior studies. Clinically, it is generally used that SUVs of 2.5 or higher is indicative as the cutoff value of cancer malignancy. Therefore, we think that using GTR using FDG SUV and FLT SUV, which are commonly used in clinical practice, will not be difficult to apply in clinical translation.
Figure 4:
- What is the % on the y-axis in B-F referring to? For example % CD3 out of what population? As CD3 is a pan-T cell marker shouldn’t CD4 and CD8 T-cells roughly add up to % CD3 (day 7 control – CD3 5 %, CD4 0.4%, CD8 0.1%)?
RE: The y-axis means the count percentage of each cell markers of total cells population from tumor. We added this in the figure legend. In order to discriminate non-lymphoid cells from lymphoid cells, the first pan-hematopoietic cell marker, CD45 positive count population, was gated. It is generally known that the sum of the absolute number or population of CD4 and CD8 cells is approximately equal to the number or population of CD3 cells. However, among the T cells expressing CD3, it is known that there are also NKT cells and DN (double negative cells, CD4-CD8-) T cells, γδ T cells [29-31]. Therefore, it is possible that other types of T cells are included.
- Figure 5:
- Please count number of positive cells (CD3, CD4, CD8 etc) on IHC stainings from several mice and show a bar diagram for every marker next to the picture. The chosen IHC area is small and the reader relies on the image selection done by the authors. An unbiased readout by cell count would strengthen the point.
RE: We thanked for your comment regarding to the count number of positive cells on IHC stainings from several mice. As a result of the number of positive cells from several mice, it was remarkably infiltrated each immune cells at the tumor site in ICI treated mice. These results are added to Figure 5 (C).
|
CD3 |
CD4 |
CD8 |
F4/80 |
|
|
Control |
8.9 ± 3.2 |
2.3 ± 0.6 |
1.2 ± 0.9 |
7.1 ± 1.4 |
|
ICI treatment |
39.7 ± 7.5 |
6.1 ± 1.6 |
15.4 ± 6.1 |
26.7 ± 6.4 |
Please also include an apoptosis marker such as cleaved Caspase3 or similar to prove the effectiveness of ICI therapy in these mice.
RE: We totally agreed with the reviewer’s comment. Although cleaved caspase3 as an apoptosis marker is also frequently identified, the effectiveness of ICI therapy can be verified through the activity of granzyme B among the most prominent serine proteases involved in the cytotoxic effect of T cells. Therefore, as a result of comparing the expression of GZB in the ICI treatment mouse tissue, it was confirmed that more GZB expression was found in the ICI treatment group than in the control
group. These results are added to Figure 5 (A).
- In the material and methods authors mention use of granzyme B in IHC staining but shown is GLUT1. Please clarify.
RE: As mentioned by reviewer’s comments above, information about Granzyme B has been clarified with the addition of GZB. We added about information of GLUT1 and Ki67 in the material and methods (Page 10, Line 351).

Reviewer 2 Report
The manuscript is well written and delivers explanations well. I only have minor comments that the authors may want to address before publishing.
Title: saying just “tumors” is quite vague. The type of tumors studied needs to be indicated.
Abstract: Please mention clearly that the research was conducted in mice, rather than “animal models”.
Fig. 1C reproduces a part of the information from Fig. 1B; it’s not necessary to duplicate data. Maybe on Fig. 1B it will be more illustrative if the two groups (drawn in red and black) are next to each other and not overlapping. Right now it’s not clear which median mark belongs to which group because both median marks are black.
Line 96: What does “establishing of mouse model” means? Was a completely new mouse model was established here? The experiment setup seems to be similar to what is described in section 2.2.
Figure 5: The scale bar font is too small to understand if it’s 10, 12 or 50, on some images? Also, it’s unreadable due to tiny letters if the scale is in millimeters (mm, probably not?) or micrometers (um, or µm). Please increase the font or add explanation to the legend.
Author Response
The manuscript is well written and delivers explanations well. I only have minor comments that the authors may want to address before publishing.
Title: saying just “tumors” is quite vague. The type of tumors studied needs to be indicated.
RE: Immune checkpoint inhibitor (ICI) treatment targeting CTLA-4 and PD-1/PD-L1 is widely used in patients with advanced melanoma, non-small cell lung cancer. It has been reported that ICI treatment has a good prognosis for patients with melanoma, and it has been correlated with improved survival [2-3]. Therefore, in this study, melanoma was targeted for ICI treatment. It was added in the introduction (Page 1-2, Line 44-46)
Abstract: Please mention clearly that the research was conducted in mice, rather than “animal models”.
RE: As commented by the reviewer, it was clearly stated that this research was conducted in “mice”. It was revised in the abstract.
Fig. 1C reproduces a part of the information from Fig. 1B; it’s not necessary to duplicate data.
RE: The reason for including this figure is to show that the various responses between the ICI therapy groups are divided from day 10 onwards, but we deleted the data because it looks like duplicate data.
Maybe on Fig. 1B it will be more illustrative if the two groups (drawn in red and black) are next to each other and not overlapping. Right now it’s not clear which median mark belongs to which group because both median marks are black.
RE: The colors have been changed in Fig. 1B to avoid confusion among readers.
Line 96: What does “establishing of mouse model” means? Was a completely new mouse model was established here? The experiment setup seems to be similar to what is described in section 2.2.
RE: We agree with the reviewer's comments. It was the same mouse model as described in section 2.2 and it was noted that this was the establishment of the mouse model. Corrected to avoid confusion among readers. The corrected part is as follows: We next acquired PET images to predict the effect of ICI treatment in mouse melanoma model. (Page 3, Line 103-104)
Figure 5: The scale bar font is too small to understand if it’s 10, 12 or 50, on some images? Also, it’s unreadable due to tiny letters if the scale is in millimeters (mm, probably not?) or micrometers (um, or µm). Please increase the font or add explanation to the legend.
RE: Thanks for the reviewer's point. The font size was enlarged and each unit was corrected to be clearly visible with a micrometer (µm).
Round 2
Reviewer 1 Report
I thank the authors for thorough revision of their manuscript.